# Ariadne’s Thread in the Developing Cerebral Cortex: Mechanisms Enabling the Guiding Role of the Radial Glia Basal Process during Neuron Migration

**DOI:** 10.3390/cells10010003

**Published:** 2020-12-22

**Authors:** Brandon L. Meyerink, Neeraj K. Tiwari, Louis-Jan Pilaz

**Affiliations:** 1Pediatrics and Rare Diseases Group, Sanford Research, Sioux Falls, SD 57104, USA; brandon.meyerink@sanfordhealth.org (B.L.M.); neeraj.tiwari@sanfordhealth.org (N.K.T.); 2Basic Biomedical Sciences, Sanford School of Medicine, University of South Dakota, Vermillion, SD 57069, USA; 3Department of Pediatrics, Sanford School of Medicine, University of South Dakota, Vermillion, SD 57069, USA

**Keywords:** neuronal migration, radial glia, cortical development, neural stem cells

## Abstract

Radial neuron migration in the developing cerebral cortex is a complex journey, starting in the germinal zones and ending in the cortical plate. In mice, migratory distances can reach several hundreds of microns, or millimeters in humans. Along the migratory path, radially migrating neurons slither through cellularly dense and complex territories before they reach their final destination in the cortical plate. This task is facilitated by radial glia, the neural stem cells of the developing cortex. Indeed, radial glia have a unique bipolar morphology, enabling them to serve as guides for neuronal migration. The key guiding structure of radial glia is the basal process, which traverses the entire thickness of the developing cortex. Neurons recognize the basal process as their guide and maintain physical interactions with this structure until the end of migration. Thus, the radial glia basal process plays a key role during radial migration. In this review, we highlight the pathways enabling neuron-basal process interactions during migration, as well as the known mechanisms regulating the morphology of the radial glia basal process. Throughout, we describe how dysregulation of these interactions and of basal process morphology can have profound effects on cortical development, and therefore lead to neurodevelopmental diseases.

## 1. Introduction

The cerebral cortex is a highly complex structure involved in higher cognitive functions such as language, reasoning, and problem solving. Importantly, most cortical neurons are generated and migrate to their final destination during embryonic stages. These processes rely on the orchestration of multiple cellular and molecular mechanisms. Disruption of any of these mechanisms can have devastating consequences and result in a vast array of neurodevelopmental disorders. For instance, it is now widely accepted that autosomal recessive primary microcephaly (a condition in which an individual is born with a smaller brain) originates from anomalies in embryonic neural progenitor survival and proliferation [1]. This conclusion holds true for microcephaly induced by genetic mutations or by environmental factors such as the Zika virus [2,3]. Emerging evidence also links autism to embryonic developmental defects. Mapping autism-linked genes to co-expression networks, two back-to-back studies concluded that autism-linked genes highly converge towards specific embryonic stages, during which neurons are generated and migrate to their final destination [4,5]. Moreover, post-mortem histological analyses of autistic children showed focal aberrant neuron organization in certain cortical areas [6]. Since the vast majority of cortical neuronal populations are generated and are topologically organized during embryonic development, these data point to neurodevelopmental deficits occurring prenatally. Many other neurodevelopmental disorders may arise, at least in part, from aberrant embryonic corticogenesis. Amongst them are lissencephaly, megalencephaly, schizophrenia, Rett Syndrome, intellectual disorders, and Down syndrome [7]. Thus, there is a need to better understand mechanisms regulating cortical development during embryonic stages. This knowledge is critical to define the etiology of neurodevelopmental disorders.

One particular cell type, radial glia, plays pivotal roles in the formation of the cerebral cortex. Radial glia not only generate neurons and glial cells through cell division, they also guide neurons migrating radially from their birthplace to their final destination. In fact, radial glia’s guiding function was demonstrated prior to their role as neural stem cells [8,9,10,11]. In this review, we focus on the cellular and molecular mechanisms that enable radial glia to serve as a guide during radial migration. Several types of radial glia exist during cortical development; however, the present review concentrates on apical radial glia residing in the ventricular zone, reporting data mostly collected in rodent models. Particular attention is brought to the function of these cells to non-cell autonomously mediate radial neuronal migration.

## 2. Radial Migration in the Cerebral Cortex

Neuron migration is an active process during which neurons sense cues in their environment to direct their migration. In response to these cues, a choreography of cytoskeletal rearrangements leads to morphological changes enabling the migration process per se. Radially migrating neurons start their migration immediately following their birth in the germinal zones lining the ventricle of the dorsal forebrain. For neurons born in the ventricular zone (VZ), the first migratory step consists of delamination from the VZ into the overlying subventricular zone [12] (SVZ, Figure 1). Following delamination, these neurons, as well as those born in the SVZ, spend a brief period in the SVZ where they assume a multipolar morphology. This switch to a transient multipolar morphology is an important step enabling slight tangential dispersion necessary to establish appropriate cortical columns circuitry [13,14]. Following this brief tangential migration, neurons migrate away from the progenitor zones in a radial fashion to populate the cortical plate (CP) (Figure 1).

During early stages, neurons use somal translocation to accomplish their migration. Somal translocating neurons present a long leading process that reaches to the pial surface. Within this leading process, their nucleus migrates into the CP to eventually reside in its final position [15]. However, at later stages (around the time when upper-layer neurons start being produced) neurons rely on glia-guided locomotion, which depends on direct interaction with radial glia [8,15]. During later developmental stages (embryonic day 14 in the mouse (E14), E65 in the macaque monkey, gestational week 12 in humans), the distance between the birthplace of neurons and the CP is too far for those neurons to extend a leading process reaching the pial surface [16,17]. Moreover, electron microscopy studies show that at later stages, migrating neurons must cross territories with high cellular density [8,17,18]. Firstly, neurons need to cross the intermediate zone (IZ), packed with axonal fibers oriented both radially and tangentially (Figure 1a). Once neurons have crossed the IZ, they need to find their way through the subplate containing Cajal–Retzius neurons and then through the CP, in which the extracellular space is extremely limited due to a high density of neurons (Figure 1a). To overcome this complex task, neurons rely on other cells to create a path and guide them through these diverse and densely populated tissues.

One of these guides is the radial glia basal process. Throughout corticogenesis, radial glia extend a basal process spanning the entire thickness of the developing cortex and ending with basal endfeet connected to the basal lamina (Figure 1). Early electron microscopy studies highlighted an intimate relationship between migrating neurons and radial glia basal processes in the cerebral cortex, suggesting a scaffolding role for radial glia during neuronal migration [8]. Such interactions were initially supported by the live imaging of cerebellar neurons migrating along glial fibers in vitro [9]. In such experiments, neurons were described as “riding the glial monorail” [19]. More recently, live imaging in organotypic brain slices confirmed the existence of glia-guided migration in the cerebral cortex [15]. At the end of radial migration, the endfeet-mediated connection of the basal process with the basal lamina is critical to prevent neurons from migrating outside the brain. Thus, like Ariadne’s thread guiding Theseus in the Minotaur’s labyrinth of ancient Greek mythology, radial glia basal processes are critical to guide migrating neurons through the labyrinth of the developing cortex (Figure 1).

## 3. Radial Glia-Dependent Mechanisms Regulating Neuron Migration

To be operational, the non-cell autonomous guiding function of radial glia requires molecular pathways enabling migrating neurons to establish and maintain their interaction with radial glia basal processes. To terminate migration, other pathways are also critical to eventually release this interaction (Figure 1b, Figure 2). All these mechanisms highly depend on receptor–ligand interactions mediated by membrane proteins in both cells.

### 3.1. Establishment and Maintenance of Neuron-Radial Glia Interactions During Migration

#### 3.1.1. Integrins

Integrins are transmembrane receptors that mediate cell-matrix and cell-cell interactions. They are obligate heterodimers composed of alpha and beta subunits, which recognize their ligands in the extracellular matrix or on other cells. In the developing mouse cortex, α3 subunits are specifically found in cells residing in the germinal zones and intermediate zone, where radial glia and newborn neurons reside. On the other hand, αV subunits are specifically observed along radial glia processes throughout the cerebral cortex [20]. Antibodies used against α3, αV, or β1 integrin subunits hamper neuronal migration in radial glia-neuron co-cultures [20]. However, while pan β1- and αV-targeting antibodies lead to the detachment of neurons from radial glia processes, α3-targeting antibodies do not [20]. This suggests that αVβ1 integrins mediate the maintenance of neuron-radial glia interactions, whereas α3β1 integrins mediate cell-cell recognition as well as cell motility mechanisms involved in migration (Figure 2). However, conditional β1 integrin Knockout (KO) in neurons does not affect their migration, while β1 integrin KO in the whole central nervous system does [21]. One explanation for this discrepancy may be that expression of β1 integrins in radial glia, but not in neurons, is essential for proper neuronal migration [21]. Thus, it would be interesting to identify the mechanisms by which β1 integrins act in radial glia to promote interactions between radial glia and neurons, and thus regulate neuronal migration.

#### 3.1.2. N-Cadherin

N-cadherin is a transmembrane protein mediating cell-cell adhesion through homophilic interactions. It is expressed in both radial glial processes and migrating neurons. N-cadherin mediated adhesion between the radial glial process and the migrating neuron is implicated in the locomotion of migrating neurons along radial glial processes [22] (Figure 2). Additionally, N-cadherin adhesion drives the polar activation of RhoA on the side of the cell with the adhesion junction, and Rac1 mediated axonal outgrowth on the contralateral side, altogether preparing neurons to transition to bipolar migration [23]. Importantly, for N-cadherin adhesion to occur between these cells, it must be expressed on the extra-cellular surface of the plasma membrane. This process is regulated by endocytic trafficking, which mediates the recycling of surface N-cadherin from the apical portion of the migrating neuron to the basal leading process. This ensures continued adhesion to the radial glial fiber as the neuron migrates radially [24]. The reduction in surface N-cadherin and associated adhesion leads to impaired progression from the multipolar to bipolar phase of neuronal migration. This results in an accumulation of neurons in the intermediate zone as the neurons are unable to migrate past the previously established neurons in the lower CP [24]. Importantly, N-cadherin is not only expressed by radial glia and migrating neurons, but also by CP neurons [22]. Therefore, once in the CP migrating neurons may not be able to distinguish surrounding neurons from radial glia basal processes, solely based on this adhesion molecule. Thus, it is probable that migrating neurons no longer rely on N-cadherin to attach to the radial glia basal process in the CP, unless repulsive cues prevent N-cadherin binding between CP neurons and migrating neurons.

Of note, the RNA binding protein FMRP mediates neuronal migration through regulation of N-cadherin expression [25]. FMRP plays many roles in neurologic function including synaptic plasticity, glial differentiation, and, fundamentally, RNA transportation and local translation [26,27]. Mutations of *FMR1*, which encodes FMRP, underlie the inherited intellectual disability Fragile X syndrome, which is the most common monogenic form of autism [26,27]. Loss of this important RNA-binding protein in mice reduces N-cadherin expression, thus affecting neuronal adhesion to radial glia. In turn, this slows multipolar migration of neurons and prolongs the multipolar to bipolar transition, delaying eventual positioning of excitatory neurons. While this does not lead to significant cortical layering aberrations in the *Fmr1* KO mice, this delay in final positioning of the neurons may contribute to neuron network aberrations and delayed circuit maturation seen in these mouse models [25,26]. These findings directly show how defects in radial migration can impair the formation of cortical circuits, and thus may lead to neurodevelopmental disease such as autism. Future studies should identify other key FMRP target RNAs that may regulate radial migration.

#### 3.1.3. Gap Junctions

The migration of neurons from the VZ also requires adhesion to the radial glia basal process via gap junctions (Figure 2). Gap junctions formed by the proteins Connexin 26 and Connexin 43 are required for the radial glia-mediated migration of neurons [28,29]. Gap junctions have the capacity to act as channels between cells; however, the adhesion property of these channels is the key characteristic mediating neuronal migration. Of note, the cytoplasmic portion of Connexin 43 does appear to have a functional impact on neuronal migration [28,29]. This implies that the connection of Connexin 43 with other cellular components plays a role in migration. In fact, the presence of Connexin 26 punctae in neurons bearing competing leading processes can predict which process is ultimately chosen by the neuron to migrate [28]. One can speculate that this is because gap junctions provide sufficient anchoring for the neurons’ cytoskeleton to proceed with migration. In migrating neurons, the accumulation and recruitment of gap junction proteins to the site of interaction are mediated by the Focal Adhesion Kinase or FAK [30] (Figure 2). FAK, encoded by the Ptk2 gene, resides in the cytoplasm where it concentrates at adhesion points between two cells or between a cell and the extracellular matrix. FAK depletion results in fewer radial glia-neuron interactions with reduced neuronal adhesion to the radial glial process [30]. Additionally, this leads to the formation of multiple highly branched leading processes. Altogether, loss of FAK results in delayed and defective migration of the neurons out of the intermediate zone and lower CP [30].

#### 3.1.4. p35

P35 is an activator of cyclin-dependent kinase 5 (Cdk5). While Cdk5 loss leads to impaired multipolar–bipolar transition [31], p35 KO leads to loss of adhesion of migrating neurons to the radial glial process [32] (Figure 2). Instead of using somal translocation or glia-guided locomotion, p35-depleted neurons migrate through another form of migration labeled as “branched migration”. This aberrant form of migration is cell-autonomous, occurs especially in the IZ, and leads to the inability of migrating neurons to move past previously migrated neurons [32]. The molecular pathway linking p35 with neuronal migration is not clear; however, work in interneurons shows that p35/Cdk5 can mediate their migration through the regulation of the ErbB4 pathway [33].

### 3.2. Termination of Neuronal Migration

#### 3.2.1. SPARC Like-1

The termination of migration leading to proper lamination of the brain during development has long been associated with the signaling molecule Reelin, which is secreted by Cajal-Retzius neurons [34]. However, neuron interactions with proteins secreted into the extracellular space by radial glia, or presented on their membrane, may also play a role by weakening neurons’ adhesion to radial glia and thus triggering the termination of migration (Figure 2). The secreted molecule SPARC like-1 (Sparcl1) seems to be expressed by radial glia, and SPARCL1 proteins are primarily localized to the upper CP [35]. Blockage of Sparcl1 function leads to the overmigration of neurons into upper cortical layers [35]. Moreover, the addition of recombinant SPARCL1 protein reduces neuronal adhesion. Finally, overexpression of Sparcl1 in apical progenitors significantly reduces migration into the cortical plate. Altogether, this suggest that radial glia may secrete SPARCL1 into the cortical plate to trigger the detachment of neurons from the basal process [35]. In the future, it will be interesting to discover the membrane receptor protein binding to SPARCL1 in neurons, and how the activation of that receptor triggers their detachment from radial glia basal processes.

#### 3.2.2. Radial Glia Endfeet-Basal Lamina Connection

In essence, radial glia are epithelial cells. Therefore, like epithelial cells in other tissues, they attach to a basal lamina, forming a boundary between the epithelium and overlying cells. In the developing brain, the basal lamina separates the brain from the meninges. Its composition includes collagens, nidogen, perlecan, and laminins [36,37] (Figure 3a). Radial glia attachment to the basal lamina occurs at the level of basal endfeet, which altogether tile the basal lamina in the manner of a jigsaw puzzle [38]. The connection between radial glia basal endfeet and the basal lamina relies on various transmembrane receptors, such as integrins and the Dystrophin-Glycoprotein Complex [37,39,40] (Figure 3a). KO or mutations of key components of the basal lamina, of various integrins, of members of the Dystrophin-Glycoprotein Complex, of the focal adhesion kinase FAK, or of the G-protein coupled receptor Gpr56 all lead to similar phenotypes [37,39,40,41,42] (Figure 3b). Pools of radial glia basal endfeet are disconnected from the basal lamina and the basal lamina presents breaches, through which neurons migrate outside of the developing cortex [37,39,40,41]. Ultimately, this ectopic neuronal migration leads to meningeal heterotopias, which are a hallmark of the neurodevelopmental defect called cobblestone lissencephaly [37,39,40,41] (Figure 3b).

Altogether, this suggests that the tiling of the basal lamina by radial glia basal endfeet constitutes a barrier preventing neurons from migrating outside the cortex. Weakened connections between radial glia and the basal lamina may lead to the detachment of basal endfeet from the lamina, which in turn primes the rupture of the basal lamina, thus enabling neurons to migrate outside of the cortex to form meningeal heterotopias. However, some questions remain. Firstly, what is the real trigger for the breakage of the basal lamina under these pathogenic conditions? Do migrating neurons digest the basal lamina due to the lack of radial glia basal endfeet protecting it? Or alternatively, is the basal lamina simply weakened by the lack of connected endfeet, which in normal conditions would help “patch” areas under stress due to the tangential expansion of the cerebral cortex? Moreover, could the disruption of the radial glia–basal lamina connection further prevent the maintenance of the basal lamina? In addition to these questions, it will be interesting to consider how key molecules mediating the attachment of radial glia to the basal lamina are delivered to basal endfeet. Related to this question, one line of research could focus on the potential importance of mRNA transport and local protein synthesis in that process, two processes known to co-exist in radial glia as described below [38,43,44,45,46].

## 4. Regulation of the Morphology of the Radial Glia Basal Process

The regulation of the morphology of the radial glia basal process is critical to ensure proper radial migration of neurons to the cortical plate. Radial glia are polarized cells. Two processes emanate from their cell body located in the VZ. The apical process extends towards the cerebral ventricle and connects with surrounding radial glia apical processes. This connection occurs at the level of the apical endfeet, containing molecular complexes involved in the regulation of cell polarity (Figure 1a). The basally oriented process connects radial glia to the overlying basement membrane. During later stages of cortical development, this basal process can become very elongated, measuring hundreds of microns in a mouse and millimeters in humans. During these stages, basal processes display dynamic extensions along their entire length, as well as complex branching in the marginal zone (MZ) situated under the pial surface [47] (Figure 1, Figure 2, Figure 3 and Figure 4). We know that the basal process can serve as a guide for neuronal migration, but what are the mechanisms controlling the morphological integrity of the radial glia basal process?

### 4.1. Generation of the Radial Glia Basal Process

The precursors of radial glia, the neuroepithelial cells, also possess a basal process (albeit much shorter). These cells divide in a symmetric fashion; therefore, both of their daughter cells present a basal process. The mechanism by which neuroepithelial basal processes are formed is controversial; it may be either through a splitting of the basal process in the mitotic mother cell [48] or through de novo growth of a new basal process in the daughter cell that did not inherit that of the mother cell [49]. Surprisingly, very little is known about how radial glia basal processes are generated and elongate. However, comprehensive live imaging of radial glia in the embryonic mouse and rat cortex shows that most if not all radial glia undergo asymmetric divisions, generating one new radial glial cell together with a differentiated cell [50,51]. These data show that following mitosis, the basal process is inherited by the cell remaining in a progenitor state. In this system, de novo growth of a basal process only occurs when radial glia are forced to undergo symmetric divisions, through the overexpression of a constitutively active form of Notch [51]. While this finding implies that radial glia devoid of a basal process retain the capacity to grow a new one, this may only occur in rare circumstances. Moreover, this suggests that elongation of the basal process may be a consequence of the expansion of the cerebral cortex and may not be dictated by molecular cues. In other words, given the known attachment of basal endfeet to the overlying basal lamina, it is possible that the basal process elongates due to the stretching forces generated by the accumulation of cells between their soma and the basal endfeet. Alternatively, the distal end of the radial glia basal processes may undergo cycles of detachment and re-attachment to the basal lamina.

### 4.2. The Morphology of the Radial Glia Basal Process Relies on a Microtubule Network

C6R cells are a radial glia-like cell-line derived from a rat glial tumor. They present processes akin to the apical and basal processes of radial glia in vivo [52]. Early work using these cells as a proxy for radial glia together with cytoskeletal inhibitors suggests that the radial glia basal process mainly relies on microtubules to maintain its overall structure [53]. In this work, the authors conclude that microfilaments may be dispensable. While this may be true for the gross morphology of the basal process, microfilaments may still be important to regulate its fine morphology. Nonetheless, the importance of microtubule regulation for radial glia morphology is further supported by the enriched expression of microtubule-related proteins in radial glia, compared to other cells [53].

Live imaging of microtubule plus-end markers in rat and mouse organotypic brain slices confirms the existence of an active microtubule network within the basal process [54,55,56] (Figure 4). These live imaging studies show that microtubule growth is largely basally directed. While the nucleation of microtubules is very active in the radial glia soma, and also particularly around the centrosome located in apical endfeet [54,56], other nucleation sites exist within the basal process [55,56]. What are those nucleation sites? Live imaging of organotypic brain slices demonstrates that microtubules can be nucleated within swellings decorating the radial glia basal process [56] (Figure 4). These swellings, also called varicosities, have been a mysterious structure until a recent study. Baffet’s group showed that these structures contain Trans Golgi Network proteins [56]. The presence of Trans Golgi Network proteins is interesting given that Golgi outposts are known microtubule nucleation sites in neurons [57]. However, certain Golgi outpost markers are absent from radial glia basal process swellings [56,58], suggesting that nucleation within the basal process differs from that found in neurons.

Nonetheless, other factors may contribute to the local nucleation of microtubules within the basal process. Amongst them are the minus-end stabilizing factors CAMSAP2 and CAMSAP3 [53,54]. Both localize at microtubule nucleation sites in the basal process (Figure 4). CAMSAP2 distribution is controlled by the Mediator of cell motility 1 (Memo1) [55], a protein present in radial glia basal processes. Anton’s group showed that Memo1 deletion leads to aberrant radial glia organization in the developing cortex [55]. Memo1-depleted radial glia show exuberant branching both along the basal process and in their distal-most region near the basal endfeet. Additionally, radial glia lacking Memo1 delaminate from the ventricular zone. These changes in radial glia morphology are associated with neuron migration deficits. Neurons fail to migrate straight to the CP, and instead adopt oblique migratory paths diverging from the typical radial orientation. These abnormal migratory paths seem to result from neurons following the exuberant branches located along the basal process of radial glia lacking *Memo1*. This suggests that supernumerary branching along the basal process may provide alternative routes for migrating neurons, and thus slow down neuronal migration (Figure 5). Given that postnatal Memo1 KO cortices display misplaced neurons within all the cortical layers, these findings suggest that slowing down neuronal migration ultimately impacts cortical cytoarchitecture. Altogether, these data confirm that an active radial glia microtubule network regulates basal process morphology, and thus plays a non-cell autonomous role in neuronal migration. Furthermore, given that *Memo1* mutations found in autistic patients also lead to a disruption of the radial glia scaffold, these findings suggest that neurodevelopmental diseases may arise from impaired radial glia morphology [55]. In the future, it will be important to further dissect the pathways mediating the basal process microtubule network, and the mechanisms utilizing that network to regulate the morphology of the basal process.

### 4.3. Radial Glia Utilize mRNA Transport and Local Protein Synthesis to Locally Deliver Key Regulators of Basal Process Morphology

Given the length of the basal process, it is important to consider the mechanisms underlying the delivery of key proteins involved in basal process morphology, thus indirectly modulating neuronal migration. In neurons, mRNA transport and local translation contribute to the regulation of their morphology [59]. Following nuclear export, mRNAs are loaded onto motor proteins to be transported to distal appendages including axons and dendrites [59]. There, various signals can trigger their translation to synthesize proteins locally and en masse. This appears to be the case in radial glia as well. Indeed, mRNAs are actively transported to the radial glia basal endfeet via the basal process [38,43,44,46] (Figure 4). In basal endfeet, these mRNAs can be utilized for local protein synthesis. One known protein mediating these mechanisms is the aforementioned RNA-binding protein FMRP [44]. FMRP is actively transported to radial glia basal endfeet, and its endfoot mRNA targets encode proteins significantly enriched for cytoskeletal regulators. These targets include kinesin molecular motors and the aforementioned microtubule minus-end binding protein CAMSAP2 [44]. Of note, mRNA transport in radial glia probably relies on the microtubules, given that the speed at which FMRP and localized mRNAs move within the basal process coincides with the known speed of microtubule-based molecular transport [59] (Figure 4).

The study of one endfoot localized mRNA candidate demonstrated that mRNA localization and local translation regulate radial glia morphology [38]. *Arhgap11a* encodes a Rho GTPase activating protein (Rho GAP), which specifically inhibits RhoA activity [60]. During cortical development, Arhgap11a expression is confined to radial glia where it localizes to the basal process of both human and mouse radial glia, in the form of mRNA and protein [38,61]. This basal process localization coincides with developmental stages during which the basal process becomes morphologically more complex, with increased branching along the basal process as well as in the basal endfeet region [38]. Knockdown of *Arhgap11a* leads to decreased branching in those two regions [38]. These morphological deficits are linked to a non-cell autonomous effect on radial migration. Wildtype neurons migrating on a radial glia scaffold depleted of *Arhgap11a* show a more tortuous migration path, leading to slower migration to the cortical plate. This impaired migration eventually leads to neuron subtypes settling in aberrant locations (Figure 5). Importantly, rescue of the branching phenotype induced by *Arhgap11a* knockdown can only occur with constructs expressing *Arhgap11a* mRNAs localizing to the basal endfeet, and capable of producing a functional ARHGAP11A protein. This finding demonstrates that basal process branching in the endfoot region relies on mRNA localization and local translation.

It is interesting to consider that neuronal migration can be impacted both by exuberant branching as found in Memo1-deficient radial glia, and by decreased branching as observed after *Arhgap11a* knockdown (Figure 5). This poses questions regarding the role of these branches. Do they provide migration cues? Could they serve as rungs of a ladder? Nonetheless, altogether these data reinforce the notion that the local regulation of RhoA is critical for radial glia basal process morphology, and thus may indirectly control the radial migration of neurons during cortical development. Moreover, it demonstrates that mRNA transport and local protein synthesis participate in the regulation of basal process morphology. However, many questions remain regarding how mRNA transport regulates basal process morphology. For instance, it is not clear whether local protein synthesis occurs solely within basal endfeet, or whether it can also take place along the basal process. Indeed, in axons and dendrites, certain moving mRNAs can show a halt correlating with local morphological remodeling, implying that local translation following that halt may participate in morphological changes [62]. This may also happen along the radial glia basal process. Moreover, could the swellings where microtubules nucleate be hot spots for local translation? In the future, it will also be interesting to consider whether mRNA transport and local translation are key to pathways mediating basal process–neuron interactions during their migration.

### 4.4. Regulation of Cell Polarity in Radial Glia

What other pathways may regulate the radial glia cytoskeleton and hence radial glia morphology and radial migration? RhoA, Cdc42, Gsk3, Apc, Numb, and MARCKS are all pleiotropic mediators of cytoskeleton dynamics and cell polarity. All of these proteins are expressed in radial glia where they localize to both apical and basal processes [63,64,65,66,67,68,69]. Knockout of any of these proteins has profound consequences on the developing cortex [64,65,66,67,68,69,70]. The depletion of either RhoA, Cdc42, Apc, Numb, Gsk3 or MARCKS disturbs apical adherens junctions, thus leading to the delamination and scattering of radial glia throughout the developing cortex. Concomitantly, proliferation appears to be impaired in radial glia [64,65,66,67,69]. These phenotypes are accompanied by a defective radial glia basal process morphology. Basal processes are misaligned, disorganized, and do not span the entire thickness of the cerebral cortex [64,65,66,67] (Figure 5). In *RhoA* KO cortices, analyses of the cytoskeleton revealed that both actin polymerization and microtubule stability are severely decreased in radial glia, whereas neurons appear normal [64]. Of note, microtubule instability was also observed in *Apc* KO cortices [68]. In *RhoA*, *Cdc42*, *Gsk3*, *Apc,* and *MARCKS* KO brains, the disturbance of radial glia morphology is accompanied by migration defects [64,65,66,67]. Interestingly, knockout of RhoA in migrating neurons on an otherwise wildtype scaffold does not disrupt their migration. However, wildtype neurons transplanted into developing cortices lacking RhoA are unable to migrate to their final destination [64]. It would be interesting to investigate whether this is also the case for Cdc42, Gsk3, Apc, Numb or MARCKS. 

Altogether, these data describe pathways regulating radial glia basal process morphology and link defective morphology with impaired neuronal migration (Figure 5). Given the pleiotropic roles of these pathways, future studies should focus on the molecular mechanisms by which these pathways influence basal process morphology. For instance, are these pathways important locally within the radial glia basal process? Or alternatively, are basal process defects downstream consequences of aberrant mechanisms occurring in the radial glia soma or in the apical endfeet? Knowing this will help us to further dissect the variety of pathways by which the morphology of the radial basal process is regulated, thus impacting radial neuronal migration. 

## 5. Conclusions and Perspectives

In this review, we laid out evidence highlighting the non-cell autonomous function of radial glia in the neuronal migration process. We focused on the known mechanisms mediating radial glia-dependent migration, and we highlighted the mechanisms that regulate the morphology of the basal process along which neurons migrate.

One major conclusion from this review is that we know very little about the local mechanisms occurring within the basal process and endfeet. This is understandable as the study of these structures is not without challenges. Tools exist to study basal process mechanisms, but most require sophisticated approaches. Live imaging of radial glia containing fluorescently labelled components in organotypic brain slices is an exquisite tool to study their dynamics within the basal process, but generating organotypic brain slices containing intact radial glia basal processes is technically challenging. However, research focused on local mechanisms occurring within basal endfeet can utilize basal endfoot preparation assays developed by the Silver group [38,43,44]. These are simple preparations as they rely on the peeling of the basal lamina from embryonic cortices using common tweezers. The power of these preparations is that they allow the isolation of basal endfeet from the cell body, while preserving some of their niche. Indeed, endfoot preparations also include the basal lamina and the overlying meninges. The downside of this method is that it does not allow the isolation of the basal process as well. Therefore, the field could highly benefit from a high-throughput in vitro model to study radial glia basal structures, including the basal process, since primary radial glia show little if any basal process in culture. However, as mentioned above, we know that radial glia are capable of generating de novo basal processes [51]. Therefore, research should focus on identifying the molecular cues guiding basal process growth. Combined with a Boyden chamber or microfluidic assays, this would give researchers the power to analyze radial glia basal structures in isolation.

To conclude, we believe that closer attention should be given to how radial glia influence neuronal migration non-cell autonomously, particularly in the context of neurodevelopmental diseases. The maintenance of the radial glia basal process, as well as neuron-radial glia interactions, may be of greater importance in higher species like humans, where the neuronal migratory path can be 10 times longer than in the common rodent models. Therefore, migration deficits observed in rodents may be more catastrophic in humans, as the human developing cortex may be more sensitive to subtle defects in radial glia morphology. Moreover, it is possible that higher species developed additional cellular mechanisms to accommodate for differences in the number and composition of the radial glia pool. Several studies suggested that this is indeed the case. In ferrets, migrating neurons switch from one radial glia basal process to another more frequently than in mice [71]. In macaque monkeys, the multipolar migratory stage appears to be absent [72]. Additionally, the importance of *Arhgap11a* in regulating radial glia basal process morphology in mice may be relevant for human brain development. During evolution, the *Arhgap11a* gene was duplicated specifically in the human lineage, which led to the emergence of the *ARHGAP11B* gene. While *ARHGAP11B* expression is critical for increased neural progenitor proliferation [73,74,75], it can also modulate Cdc42 activity [60], which is known to regulate radial glia basal process morphology [66]. Therefore, it is possible that *ARHGAP11B* also mediates radial glia basal process morphology, and thus may be important for neuron migration. Finally, future studies should evaluate whether the mechanisms identified in the apical radial glia within the VZ (aRG) are transferable to outer radial glia (oRG), which play a major role during cortical development in higher species [16,72]. oRG are structurally and molecularly different from their aRG counterparts [16,76,77], hence it is likely that distinct mechanisms regulate oRG- vs. aRG-mediated neuronal migration. Thanks to brain organoid technology [78], we now have the capacity to address these questions in the context of human development.

## Figures and Tables

**Figure 1 cells-10-00003-f001:**
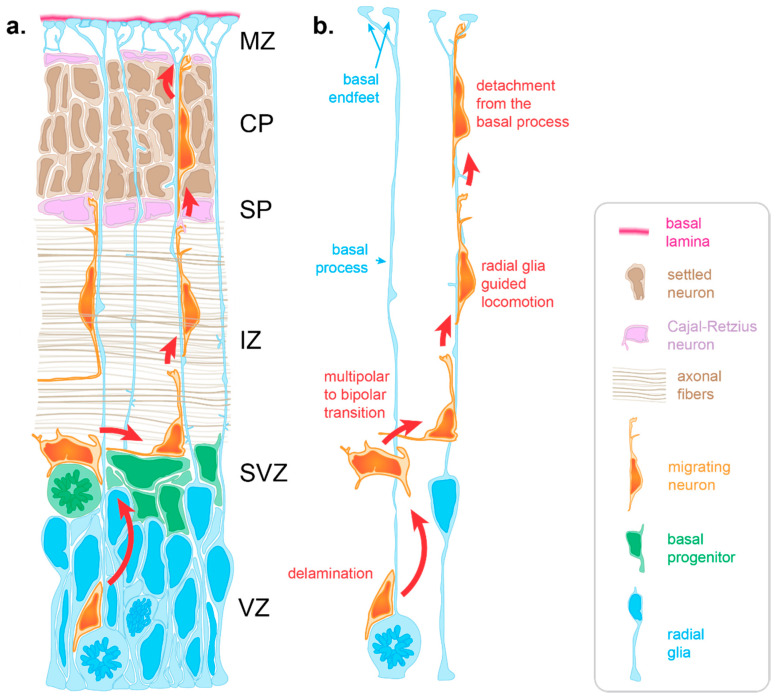
Radial migration in the developing cortex. (**a**) Radial migration relies on the radial glia basal process for guidance. Immature neurons born at the apical surface of the developing cortex use the basal process to navigate through varied and densely packed histological layers before resting in their final location within the cortical plate (CP). (**b**) Neuronal migration occurs in several key stages. Delamination from the ventricular zone (VZ) brings the newly differentiating neuron to the subventricular zone (SVZ), where it adopts a multipolar morphology allowing for brief tangential migration. Recognition of the radial glia basal process by the migrating neuron drives a transition to a bipolar morphology as basal process–neuron interactions enable radial glia to guide radial migration. Glia guided locomotion is necessary for the neuron to negotiate the compact layers of settled cortical neurons. Once the neuron is properly positioned, termination signaling leads to detachment from the basal process. (IZ: intermediate zone, SP: subplate, MZ: marginal zone).

**Figure 2 cells-10-00003-f002:**
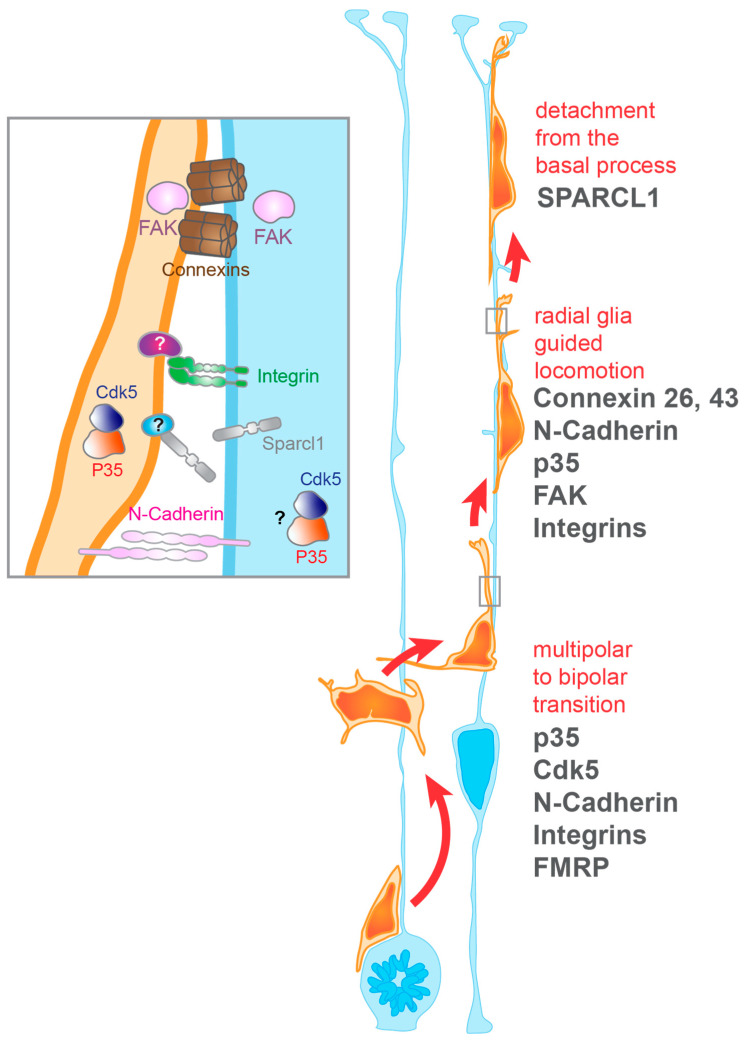
Mechanisms regulating radial glia-migrating neuron interactions involved in specific migratory stages. Inlay shows the cell surface interface between the radial glia process and the migrating neuron. Cell surface proteins and their regulation are critical to the proper guidance of the migrating neuron.

**Figure 3 cells-10-00003-f003:**
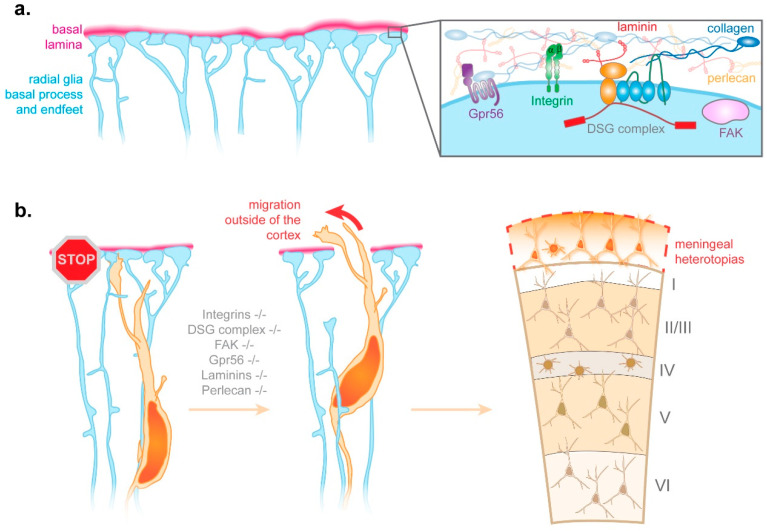
Importance of the radial glia basal endfoot-basal lamina connection for radial migration. (**a**) The basal portion of the radial glial process binds to the extracellular matrix of the basal lamina through transmembrane adhesion receptors located at the level of so-called basal endfeet. Collectively, these endfeet form a barrier demarking the basal boundary of the developing cortex. (**b**) The endfoot barrier restricts neuronal migration to the cortical plate. Compromise of this barrier through malformation of the basal lamina or aberrant endfoot adhesion can lead to neuronal overmigration and ectopic placement in meningeal layers. (DSG complex: Dystrophin-Glycoprotein Complex, FAK: Focal Adhesion Kinase).

**Figure 4 cells-10-00003-f004:**
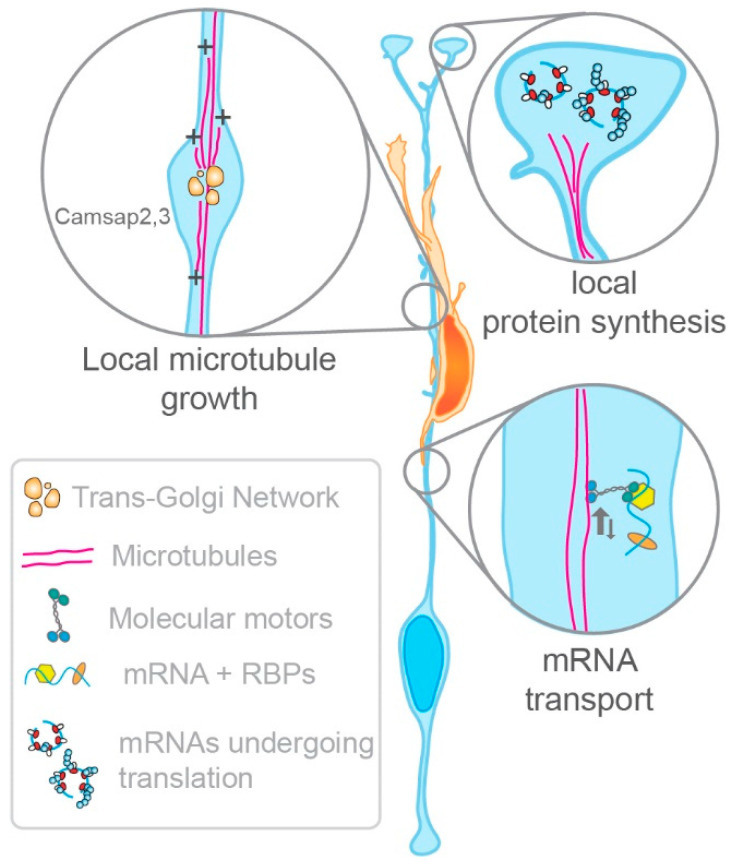
Basic mechanisms involved in the regulation of the morphology of the basal process. Left: A microtubule network regulates the radial glial basal process morphology and thus facilitates radial migration. Microtubule nucleation occurs locally within the radial glia basal process. Known proteins that help in the nucleation of microtubules are localized in varicosities containing Trans Golgi Network proteins as well as the minus-end binding proteins CAMSAP2 and CAMSAP3. Right: mRNAs regulating the shape of the radial glia basal are transported within the basal process and locally translated in the basal endfeet (RBP: RNA-Binding protein).

**Figure 5 cells-10-00003-f005:**
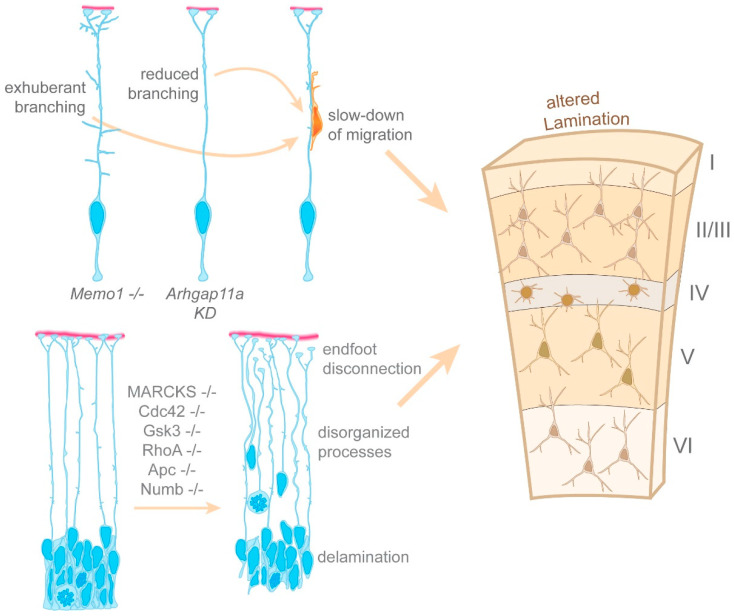
Pathways regulating radial glia morphology and the shape of the basal process. *Memo1* and *Arhgap11a* regulate branching of the radial glia basal process. Knockout of *Memo1* and knockdown of *Arhgap11a* lead to exuberant, and decreased branching of the basal process, respectively. Ultimately, compromised basal process branching slows down neuronal migration and leads to misplaced neurons in the cortex. The depletion of proteins such as RhoA, Cdc42, Gsk3, Apc, Numb and MARCKS leads to radial glia endfoot disconnection, the disorganization of basal processes, and radial glia delamination from the apical surface. In these conditions, impaired radial glia morphology is linked with altered lamination of the cortex.

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
