# Peer review of "Ariadne’s Thread in the Developing Cerebral Cortex: Mechanisms Enabling the Guiding Role of the Radial Glia Basal Process during Neuron Migration"

_cells, 2020, doi:10.3390/cells10010003_

Round 1

Reviewer 1 Report

This is a concise and focused review of factors potentially affecting radial glial cells' guidance of their progeny in the developing cortex. The figures are generally well designed and consistent, with some suggestions for added content given below. This will be a useful resource for the field. The majority of my suggestions are minor changes for clarity, with only one larger scientific question: It is surprising to this reader, especially given the discussion of ARHGAP11A, that almost no consideration is given to human or primate-specific features affecting cortical size and lamination. What about ARHGAP11B? How might expression of additional components affect radial glia morphology, cytoarchitecture, or adhesion proteins? An additional paragraph or section on variation across species would be valuable.

General stylistic comment: The MS overuses the word “indeed” and the phrase “plays/holds a role” (the latter is used 18 times in the body of the review) – the first could simply be omitted for clarity and flow, for the second consider using more specific language. Is the component required? Necessary? Implicated?

Figure 1 – what is the bright pink at top (presumably meninges), and what are the fibers crossing the IZ? These should also be included in the legend. “delamination,” since it applies to the migrating neurons, should perhaps be on the left side of the panel under “multipolar to bipolar transition.” IZ is also never defined in the legend.

Line 73/74 – could the authors include some specific numbers on “later stages” – what embryonic days / gestational weeks are these?

Line 83 – “such” is subjective and should perhaps be replaced by “these”

Line 94 – Minotaur’s labyrinth OF ancient Greek mythology?

Line 114 – first instance of “proteins” can probably be deleted for clarity.

Lines 134/135 – this sentence is somewhat unclear – “on the side of the junction” should perhaps be “on the face of the cell containing the junction” or similar

Line 154 – suggest omitting the phrase “are very interesting as they…” – simply describe what they show and let the reader decide they are interesting.

Figure 2 – it is a nice visual through-line to use and build on the components of Figure 1, but could more be added to illustrate the structure of some of the cell-cell adhesion complexes discussed, as in Figure 3?

Lines 267-269 – incorrect grammar in sentence, recraft fo clarity

Line 276 – What are C6R cells? What species? Please elaborate.

Line 309-310 – correct the grammar here; glia are plural but sentence uses singular

Lines 355-358 – somewhat unclear; recraft sentence?

Line 422 – why is live imaging technically challenging? (this may well be true, but it’s not clear to a novice reader why this is any more or less technically challenging than endfeet preparation described in following sentence).

Line 423 – throughout the rest of the review, only the author’s last name is used; suggest the same here – “the Silver group”

Line 432-433 – only “Boyden” should be capitalized here?

Author Response

We thank the reviewer for his constructive comments. In the following point-by-point response, the reviewer’s comments are italicized, and our responses are in blue. The lines we refer to are those from the manuscript we submitted after revision, and using “track changes” in Word.

This is a concise and focused review of factors potentially affecting radial glial cells' guidance of their progeny in the developing cortex. The figures are generally well designed and consistent, with some suggestions for added content given below. This will be a useful resource for the field. The majority of my suggestions are minor changes for clarity, with only one larger scientific question: It is surprising to this reader, especially given the discussion of ARHGAP11A, that almost no consideration is given to human or primate-specific features affecting cortical size and lamination. What about ARHGAP11B? How might expression of additional components affect radial glia morphology, cytoarchitecture, or adhesion proteins? An additional paragraph or section on variation across species would be valuable.

We thank the reviewer for this comment. It would indeed be valuable to develop on our considerations related to higher species. Therefore, we added several sentences in the very last paragraph of this review (line 464-479), to discuss recent data collected in Ferrets, macaque monkeys, and also related to how ARHGAP11B might play a role in the mechanisms described in this review.

General stylistic comment: The MS overuses the word “indeed” and the phrase “plays/holds a role” (the latter is used 18 times in the body of the review) – the first could simply be omitted for clarity and flow, for the second consider using more specific language. Is the component required? Necessary? Implicated?

This is a helpful comment that we will take into account not only for the present manuscript, but also for any other manuscript moving forward. We removed the word “indeed” in the following instances: lines 53, 79, 90, 235, 317, 327, 355, 369, 400, 440, and 451, and we have changed the phrase “plays/hold a role” here: lines 59, 142, 158, 175, and 433.

Figure 1 – what is the bright pink at top (presumably meninges), and what are the fibers crossing the IZ? These should also be included in the legend. “delamination,” since it applies to the migrating neurons, should perhaps be on the left side of the panel under “multipolar to bipolar transition.” IZ is also never defined in the legend.

We thank the reviewer for these suggestions, which will be helpful to the reader. We added the bright pink line and the IZ fibers to the legend. We also moved “delamination” to the left, and we defined IZ in the figure legend.

Line 73/74 – could the authors include some specific numbers on “later stages” – what embryonic days / gestational weeks are these?

Line 80 - We added these details between parentheses, together with citations that support these time points.

Line 83 – “such” is subjective and should perhaps be replaced by “these”

Line 88-89 – We replaced “such” by “this” and “these” as suggested by the reviewer.

Line 94 – Minotaur’s labyrinth OF ancient Greek mythology?

Line 100 – We agree with the reviewer and we implemented this change.

Line 114 – first instance of “proteins” can probably be deleted for clarity.

Line 120 – We removed the first instance of “proteins” as suggested.

Lines 134/135 – this sentence is somewhat unclear – “on the side of the junction” should perhaps be “on the face of the cell containing the junction” or similar

We agree that this sentence was unclear. We modified it. Please see the new sentence, lines 133-134.

Line 154 – suggest omitting the phrase “are very interesting as they…” – simply describe what they show and let the reader decide they are interesting.

Line 158 – we agree with the reviewer and we deleted “are very interesting as they…”.

Figure 2 – it is a nice visual through-line to use and build on the components of Figure 1, but could more be added to illustrate the structure of some of the cell-cell adhesion complexes discussed, as in Figure 3?

This is a good point, and will be useful to the reader. We modified Figure 2 to illustrate the key players described in the manuscript.

Lines 267-269 – incorrect grammar in sentence, recraft fo clarity

Line 284 – We modified this sentence.

Line 276 – What are C6R cells? What species? Please elaborate.

Lines 292, 293 - We added a sentence describing those cells, together with the reference of the article describing their generation.

Line 309-310 – correct the grammar here; glia are plural but sentence uses singular

Line 328 – We thank the reviewer for catching this typo, we corrected it.

Lines 355-358 – somewhat unclear; recraft sentence?

Lines 372-375 – We agree that the sentence was convoluted, we changed it.

Line 422 – why is live imaging technically challenging? (this may well be true, but it’s not clear to a novice reader why this is any more or less technically challenging than endfeet preparation described in following sentence).

We agree that the “however” we used to connect the two sentences was not warranted given the lack of explanation regarding the difficulty of generating organotypic brain slices. We added: “generating organotypic brain slices containing intact radial glia basal process is technically challenging”, line 442.

Line 423 – throughout the rest of the review, only the author’s last name is used; suggest the same here – “the Silver group”

Line 444 – This is a good point, we modified as suggested.

Line 432-433 – only “Boyden” should be capitalized here?

Line 454 – We agree with the reviewer and we corrected as suggested.

Reviewer 2 Report

This is a very important, timely and a very well written review about cortical radial glia cells and their relevance for guiding the migration of cortical neurons, also in the context of associated diseases. Moreover, the review contains illustrations that nicely support understanding.

The review is well structured and collects the most relevant literature of the topic in a clear and informative manner.

I have only few remarks.

In the abstract, the authors mention human and mice, and in the introduction related human diseases are discussed. However, when it comes to paragraph 2, it is not clearly indicated that the authors here only refer to mice (e.g. in matters of neurogenesis). This should be done. Further, the comparison to the processes in the human brain are worth to discuss as well, especially as at the end in line 444 the outer glia cells of the human brain are mentioned by the authors.

When the authors discuss in paragraph 3 the mechanisms that underlie glia-dependent migration, such as integrins and connexins, I wonder what exactly makes the difference? What makes the migrating neurons "stick" to the radial glia and not e.g. to the neurons within the cortical plate, which the migrating cells have to pass by. Is N-Cadherin for example only expressed by radial glia and migrating neurons and NOT by post mitotic neurons of the cortical plate? Or are there other rather repellent cues expressed by cortical plate neurons so that migrating cells preferentially interact with RGCs. This could be highlighted in more detail in my opinion.

Author Response

We thank the reviewer for his/her constructive comments. In the following point-by-point response, the reviewer’s comments are italicized, and our responses are in blue. The lines we refer to are those from the manuscript we submitted after revision, and using “track changes” in Word.

This is a very important, timely and a very well written review about cortical radial glia cells and their relevance for guiding the migration of cortical neurons, also in the context of associated diseases. Moreover, the review contains illustrations that nicely support understanding.

The review is well structured and collects the most relevant literature of the topic in a clear and informative manner.

I have only few remarks.

In the abstract, the authors mention human and mice, and in the introduction related human diseases are discussed. However, when it comes to paragraph 2, it is not clearly indicated that the authors here only refer to mice (e.g. in matters of neurogenesis). This should be done.

We thank the reviewer for this comment. In paragraph 2, we actually meant to be general and not only describe neurogenesis in the mouse. The mechanisms we describe are also present in higher species, but we did not want to add additional processes occurring in higher species that would not be discussed in the rest of the manuscript (although we do allude to them at the very end of the manuscript). With that being said, we tried to address the reviewer’s comment by adding these to the manuscript:

  1. In the introduction we added “reporting data mostly collected in rodent models” to set the stage for the next paragraph described neurogenesis.
  2. As suggested by reviewer 1, we added the stages that we described as “later stages” line 80. We did this for the mouse, the macaque monkey and humans.

Further, the comparison to the processes in the human brain are worth to discuss as well, especially as at the end in line 444 the outer glia cells of the human brain are mentioned by the authors.

This is a good comment that was also mentioned by reviewer 1. To address this, we added content to the very last paragraph of the manuscript describing implications to migration in higher species (lines 463-478).

When the authors discuss in paragraph 3 the mechanisms that underlie glia-dependent migration, such as integrins and connexins, I wonder what exactly makes the difference? What makes the migrating neurons "stick" to the radial glia and not e.g. to the neurons within the cortical plate, which the migrating cells have to pass by. Is N-Cadherin for example only expressed by radial glia and migrating neurons and NOT by post mitotic neurons of the cortical plate? Or are there other rather repellent cues expressed by cortical plate neurons so that migrating cells preferentially interact with RGCs. This could be highlighted in more detail in my opinion.

We thank the reviewer for this very insightful comment. Regarding Integrins, we know that their expression is not relevant to radial glia-guided migration based on data by the Muller lab described in the manuscript. Therefore, it is likely that their presence only in radial glia is critical for cortical development. However, to our knowledge we do not know what is the neuronal ligand mediating this mechanism. We highlighted this in the new version of figure 2, and we also mentioned the immunohistochemistry data showing Integrin expression in the developing cortex (lines 123-127). Regarding N-Cadherin, it is harder to interpret, because radial glia, migrating neurons and CP neurons do all express it, as mentioned by the reviewer and by immunohistochemical analyses. Thus, we added sentences describing this dilemma, together with the interpretation provided by the reviewer (lines 152-156).